# Potential Anticancer Activities and Catalytic Oxidation Efficiency of Platinum(IV) Complex

**DOI:** 10.3390/molecules27144406

**Published:** 2022-07-09

**Authors:** Mohamed M. El-bendary, Tamer S. Saleh, Mansour M. Alomari, Ehab M. M. Ali, Bambar Davaasuren, Mariusz Jaremko, Bandar A. Babgi

**Affiliations:** 1Department of Chemistry, College of Science, University of Jeddah, Jeddah 21589, Saudi Arabia; tssayed@uj.edu.sa (T.S.S.); 1900066@uj.edu.sa (M.M.A.); 2Chemistry Department, Faculty of Science, Tanta University, Tanta 31527, Egypt; 3Green Chemistry Department, National Research Centre, Dokki, Giza 12622, Egypt; 4Biochemistry Department, Faculty of Science, King Abdulaziz University, Jeddah 21589, Saudi Arabia; emali@kau.edu.sa; 5Division of Biochemistry, Department of Chemistry, Faculty of Science, Tanta University, Tanta 31527, Egypt; 6Core Labs, King Abdullah University of Science and Technology (KAUST), Thuwal 23955-6900, Saudi Arabia; bambar.davaasuren@kaust.edu.sa; 7Biological and Environmental Science and Engineering (BESE) Division, King Abdullah University of Science and Technology (KAUST), Thuwal 23955-6900, Saudi Arabia; mariusz.jaremko@kaust.edu.sa; 8Department of Chemistry, Faculty of Science, King Abdulaziz University, P.O. Box 80203, Jeddah 21589, Saudi Arabia; bbabgi@kau.edu.sa

**Keywords:** platinum(IV), Pyridine-2-carbaldehyde-oxime, anticancer agent, catalytic oxidation activity, luminescence

## Abstract

The treatment of an aqueous acetonitrile solution of chloroplatinic acid hydrate H_2_PtCl_6_.xH_2_O and pyridine-2-carbaldehyde-oxime (paOH) in the presence of potassium thiocyanate at room temperature (25°) led to the formation of a new Pt(IV) complex with the formula **[Pt(SCN)_2_(paO)_2_], (1).** Complex **1** was fully characterized by FT-IR, UV-vis and NMR spectroscopic techniques as well as elemental analysis. The crystallographic structure of complex **1** was obtained by single-crystal X-ray diffraction. The structure of complex **1** consists of a distorted octahedral geometrical environment around the platinum center in which the coordination sites are occupied by two terminal thiocyanate ligands in *trans* arrangement and two bidentate paO ligands through four nitrogen atoms. In addition, the in vitro evaluation of the cytotoxicity of platinum complex **1** against four different cancer cell lines was performed. The IC_50_ values for colon (HCT116), liver (HepG2), breast (MCF-7) and erythroid (JK-1) treated with complex **1** are 19 ± 6, 21 ± 5, 22 ± 6, and 13 ± 3 μM, respectively. In HCT116 cells treated with the IC50 dose of our title compound, apoptosis and necrosis were increased by 34% and 27.8%, respectively. Cells halted in the proliferative phase (S phase) to 21.7 % and 29.8% in HCT116 and HepG2 cells treated with complex **1** have anti-proliferative actions. Furthermore, the catalytic activity of synthesized complex **1** was examined in the oxidation reaction of benzyl alcohols in the presence of an oxidant. Finally, the luminescence behavior of complex **1** was investigated.

## 1. Introduction

Platinum(II)-based complexes such as cisplatin, oxaliplatin and carboplatin have been acknowledged as a practical class of anticancer drugs utilized in treating various types of cancer. The most well-known conventional platinum anticancer complex is cisplatin (*cis-*diamminedichloroplatinum, which effectively treats testicular, ovarian, head, neck and small cell lung cancer [1,2]. The chemical structures consist of two *cis*-oriented inert ammine ligands in combination with two relatively-labile chloro ligands or oxygen-donating chelate ligands; the four donating atoms are arranged in a square planar environment around the metal [3]. Their mechanism of action is believed to be through covalent binding to DNA, changing the double helical structure and triggering cell death [4]. However, the shortcomings associated with Pt(II)-based drugs such as drug resistance and severe side effects motivated the development of new platinum based drugs to overcome or reduce these drawbacks [5,6]. In this domain, a new trend entails exploring and developing multi-action Pt(IV) complexes with potential anticancer activities [7,8]. The octahedral Pt(IV) complexes can be seen as the original square planar Pt(II) compounds with two extra ligands in the axial positions. Although octahedral low-spin d^6^-metal is quite inert to ligand substitution, the reduction can be facilitated to produce the original Pt(II) drug and the free two axial ligands which make Pt(IV) good prodrugs that can be activated inside the cancer cells [9]. The choice of the axial ligands is important in controlling the activation and pharmacological features of the prodrugs. Most reported Pt(IV) prodrugs are structurally-related to cisplatin (using the cis-PtCl_2_(NH_3_)_2_ with two extra axial ligands). Axial ligands can be biologically-inactive such as hydroxido or halo ligands [10,11] or biologically active ligands [12,13,14,15,16]. The compound *cis, trans*-[PtCl_2_(OOCCH_3_)_2_(NH_3_)(NH_2_Cy)] (known as satraplatin) exhibits a promising toxicity profile against breast, prostate and lung cancer [17] and was progressed to phase III clinical trial against hormone-refractory prostate cancer (PC); however, there was no overall survival benefit. Other Pt(IV) compounds have been clinically trialed such as ormaplatin and iproplatin [18].

The oxidation of primary and secondary alcohols into the corresponding carbonyl compounds plays a central role in organic synthesis [19,20,21]. Traditional methods for performing such transformations generally involve the use of chromium(VI) oxidants [22]. However, from viewpoint of avoiding environmental problems, catalytic efficient methods that employ clean oxidants such as O_2_ and H_2_O_2_ were investigated. Recently, oxidation of alcohols in the presence of oxidant catalyzed via transition metal has been established [23,24,25]. Most benzyl alcohol derivatives acquire an economic value when oxidized to aldehydes, either because aldehydes are known intermediates to high-value components, or because they have a very high market value themselves, being widely used in perfume, cosmetics and food industries [26]. The effective catalytic conversion of benzyl alcohol derivatives requires the study of the relation between the substrate features and the catalytic behavior at the molecular level. The correlation between the catalyst’s structural chemistry and the characteristics of the reactant in a specific environment allows development of rules for successful catalyst design. Indeed, among the factors that strongly influence catalytic activity, the substrate effect can play a crucial role. Pt complexes are an environmental-friendly alternative for the oxidation of alcohols in base-medium, although the main product is often the corresponding monoacid and not the desired aldehyde.

From these points aforementioned, in the current work, we sought to investigate the anticancer properties of newly synthesized Pt(IV) with axial thiocyanate ligands and two chelating pyridyl-oxime ligands. Moreover, we decided to check the catalytic activity of the new synthesized Pt(IV) complex **1** for the oxidation reaction of benzyl alcohols in the presence of hydrogen peroxide as an oxidant. The main research goal is to study the potential anticancer activities and catalytic oxidation efficiency of platinum(IV) complex **1**.

## 2. Materials and Methods

### 2.1. Chemicals and Reagents

All reagents, chemicals and organic solvents were bought from commercial sources and utilized as received unless otherwise noted. All additional chemicals were acquired without further purification from Merck (Darmstadt, Germany), Sigma-Aldrich (St. Louis, MO, USA), or Acros Organics (part of Thermo Fisher Scientific, Waltham, MA, USA)

### 2.2. Instrumentation

Thin-layer chromatography was done on Merck 60 GF254 silica gel plates that had been precoated with a fluorescent indicator, and detection was done with UV irradiation at 254 and 360 nm. The melting points were determined using Stuart melting point equipment without any adjustments. On the Nicolet iS10 FT-IR spectrometer, IR spectra were captured using a Smart iTR, which is an ultrahigh-performance, flexible attenuated total reflectance sampling attachment (Thermo Fisher Scientific). A Bruker Avance III 400 (9.4 T, 400.13 MHz for ^1^H, 100.62 MHz for ^13^C) spectrometer (Bruker, Billerica, MA, USA) with a 5-mm BBFO probe was used to record NMR spectra at 298 K. Chemical shifts (δ in ppm) are relative to the internal standards. On a EuroVector C, H, N, and S analyzer, elemental analyses were performed (EA3000 series). Shimadzu (UV-310l PC) spectrometer was used to record electronic absorption spectra. The Cary Eclipse Fluorescence Spectrophotometer (λ_ex_ = 290 nm) was used to measure fluorescent spectra. Elma Sonicator P30H device with the ultrasonic frequency of 37 kHz and power of 320 W was used for sonication (max.). After 25 min of operation, the bath temperature was raised from 25 to 70 degrees Celsius. All of the reactions took place at 60–70 degrees Celsius, which was maintained by adding or removing water in an ultrasonic bath (the temperature inside the reaction vessel was 66–68 °C). The general procedure for the oxidation of benzyl alcohols is given in the Appendix A.

### 2.3. Synthesis of **[Pt(SCN)_2_(paO)_2_]** (1)

Complex **1** was prepared at room temperature (25°) by self-assembly between an aqueous solution of chloroplatinic acid hydrate H_2_PtCl_6_.xH_2_O (0.102 g, 0.25 mmol) and a mixture solution of pyridine-2-carbaldehyde-oxime (paOH) (0.5 mmol, 0.054 g) in 25 mL acetonitrile and potassium thiocyanate (0.097 g, 1 mmol) in 10 mL H_2_O (Figure 1). The yellow solution mixture was stirred at room temperature for 30 min, and then the solution was kept enclosed at room temperature. After slow evaporation, red prismatic crystals were obtained after ten days. The crystals were filtered off, washed with mixed solvents of CH_3_CN and H_2_O and were air-dried. Complex **1** was produced with a yield of 85 mg (61%). Elemental Analysis. Anal. Calc. (%) for C_14_H_10_N_6_O_2_PtS_2_, **1**: Calcd. C, 30.38; H, 1.82; N, 15.18; Found (%): C, 30.25%; H, 1.91; N, 15.10. IR, KBr (cm^−1^) (v, very; s, strong; m, medium; w, weak): 3048 w, 2925 m, 2126 s, 1607 s, 1512 s, 1485 s, 1429 s, 1358 m, 1320 m, 1253s, 1188 s, 1155 m, 1115 m, 1049 m, 1034 w, 898 m, 819 s, 769 s, 738 m, 676 s, 567w, 544 w. ^1^H NMR (DMSO-d6, ppm): δ = 7.16 (dd, 1H, H_b_), 7.61 (d, 1H, H_d_), 7.73 (s, 1H, HC=N), 7.61 (dd, 1H, H_c_), 8.48 (d, 1H, H_a_). ^13^C NMR (400 MHz, DMSO-d6, ppm): δ = 119.1 (C_b_), 120.5 (C_d_), 140.5 (C=N), 140.9 (Cc), 147.0 (Ce), 157.8 (C_a_) 195.9 (SCN).

### 2.4. Crystallographic Studies

Crystals were obtained by slow evaporation of the complex solution in water and acetonitrile. The collection of single crystal X-ray diffraction data for the complex was carried out on a Bruker D8 Venture single crystal X-ray diffractometer at 120 K, using Mo Kα (λ = 0.71073) radiation. The crystal structure was solved with the SHELXT [27] structure solution program and refined with the SHELXL [27] implemented in Olex2 [28] program package. The crystal structure refinement details for C_14_H_10_N_6_O_2_PtS_2_ are listed in Appendix A. Full crystallographic data can be found under the deposition numbers (CCDC): 2096540, 12 Union Road, Cambridge CB21 EZ, UK (e-mail: datarequest@ccdc.cam.ac.uk).

### 2.5. Assessments of IC_50_ of Complex **1** in HCT_116_, HepG2, MCF-7 Cell and JK-1 Lines by Using xMTT Assay

Tissue Culture Unit, Department of Biochemistry, Faculty of Science, King Abdulaziz University provided 4 human cell lines (3 attached and one suspended) colon (HCT_116_), liver (HepG2), breast (MCF-7) and erythroid (JK-1). Attached human cell lines were cultured for 24 h in a flask containing complete media Dulbecco’s Modified Eagle’s Medium (DMEM). JK human cell lines were cultivated in Roswell Park Memorial Institute Medium (RPMI 1640), which contains 10% fetal bovine serum and 1% antibiotic, at 37 °C and 95% humidity in a sterile 5% CO_2_ incubator. DMEM and RPMI 1640 were purchased from Life Technologies Gibco. 4 mL of 0.25 percent trypsin with EDTA was given to the individual cells after 90 % of the confluent cells were collected and incubated in a CO_2_ incubator for 5 min. With the addition of 5 mL of complete medium, the trypsin reaction was stopped. After centrifugation of the unattached cells containing the medium, the pellets were washed twice with sterile phosphate buffer saline (PBS) [29].

The number of cells was determined with a hemocytometer and counted in the four major squares after 20 μL of this cell-containing media was stained with 0.4 percent trypan blue. By multiplying the counted cells by 1/4 × 10^4^ × 2, the number of cells per mL was calculated. 0.1 mL of 5000 cells suspended in complete medium per well were put on a 96-well microplate, and the plate was incubated in the incubator for 24 h.

Different concentrations of platinum complex **1** (3.5, 7, 14, 28, 56 μM) were added to the media when 70% of the cells in each well were confluent. Each concentration was repeated four times. For 48 h, the plate was incubated.

In each well, the medium was replaced with 100 μL of 0.5 mg MTT/mL free medium for 4 h in the incubator. Each well was filled with 100 μL dimethyl sulfoxide (DMSO) and incubated at room temperature for 15 min before being monitored using a microplate reader at 595 nm (Bio-RAD microplate reader). 50% of an inhibitory concentration of IC_50_ of Pt(SCN)_2_(PaO)_2_ treated with 3 attached cell lines and JK-1 suspended cell line were calculated from the curve of the % percentage of cell viability versus different concentration of platinum IV complex using GraphPad prism version 9 software [30].

### 2.6. Determination of Apoptosis and Necrosis by Using Annexin V-FITC/Propidium Iodide (PI) in HCT_116_, HepG2 Treated with Complex **1**

HCT_116_ and HepG2 were transplanted and cultured for 24 h in a CO_2_ incubator. The cells were separated by trypsin and counted. 2 × 10^5^ cells were cultivated and cultured for 24 h in a 6 well plate. The medium in the well was replaced with complete medium containing IC_50_ of complex **1**. The morphology of HCT_116_ and HepG2 were seen under inverted microscope.

After 24 h, the HCT_116_ and HepG2 cells were separated and the medium of each well-containing cell was removed into tubes and centrifuged. Following that, the pellets were washed in phosphate buffer solution (PPS). The cells were suspended in the liquid. 100 μl of treated HCT_116_ or HepG-2 cells and 25 μl Annexin V-FITC/propidium iodide (PI) solution were placed in an Eppendorf tube. The tubes were incubated for 15 min in the dark. After the incubation period, 400 μL of binding buffer was added. Fluorescence is a color that glows. A flow cytometry instrument was used to detect the cells. The software module does calculations automatically [31].

### 2.7. Flow Cytometry Analysis of the Cell Cycle of HCT_116_ and HepG2 Treated with [Pt(SCN)_2_(PaO)_2_] Using the Propidium Iodide Technique

Propidium iodide (PI) (from ThermoFisher Scientific, Waltham, MA, USA) can bind to DNA and stain it, as well as quantify cellular aggregation in each phase of the cell cycle using flow cytometry [32]. 1 × 10^6^ HCT_116_ or HepG2 cells were seeded into a 6-well plate for 24 h. The medium was replaced with complex **1** at the IC_50_ level. After 24 h of incubation, treated HepG2 cells were collected by adding 0.5 mL 0.25 percent trypsin and terminating the activity of trypsin by adding 0.5 mL complete medium in each well. After that, the suspended cells were centrifuged for 5 min at 1500 rpm and washed twice with PBS. The cells were fixed with 1 mL of ice-cold 70% ethanol for at least 4 h at −20 °C. The 100 μL of washed suspended cells with cold PBS with RNase A were then stained with 250μL of PI solution (50 mg/mL PI) and incubated in the dark for 1 h. All of the tagged cells were read using a flow cytometer (Applied Bio-system, Waltham, MA, USA).

## 3. Results and Discussion

### 3.1. Crystal Structure Description of **[Pt(SCN)_2_(paO)_2_] (1)**

Reaction between platinic acid and pyridine-2-carbaldehyde-oxime in the presence of potassium thiocyanate in aqueous acetonitrile solution leads to the formation of the complex with the formula; **[Pt(SCN)_2_(paO)_2_], (1).** The asymmetric unit of complex **1** comprises a half formula unit, which contains one platinum atom with half occupancy, one of pyridine-2-carbaldehyde-oxime (paO) ligand with deprotonated oxime group and one thiocyanate group. Two repeat asymmetric units represent the ORTEP diagram of the molecular structure of complex **1**, which consists of one crystallographically independent platinum, two paO ligands and two terminal thiocyanate groups as shown in Figure 1. The centrosymmetric Pt^IV^ complex is formed via coordination of the central Pt atom by two axial S-bound thiocyanate anionic ligands in a *trans* configuration and two pyridine-2-carbaldehyde-oxime ligands chelating through their N atoms in equatorial sites. The Pt atom is coordinated to pyridine-2-carbaldehyde-oxime ligand (paO) forming a five-membered ring through Pt1N2C7C6N3 (Figure 1). The structure of complex **1** consists of distorted octahedral coordinated platinum atoms in which the coordination sites are occupied by two terminal thiocyanate ligands in *trans* arrangement and two bidentate paO ligands through four nitrogen atoms. Selected bond lengths and angles are listed in Appendix A. Complex **1** is air and light stable and is freely soluble in DMF and DMSO solvents. In complex **1**, the bidentate ligand lost the hydrogen of the oxime group, adopting the anionic form. The N-O bond length is 1.2678(8), which is approaching the bond length seen in nitro groups; whereas the bond length of C-N_oxime_ is 1.3124(9), which is in the same range observed for C-N bond lengths in pyridines. This suggests that the negative charge is delocalized over the whole ligand and stabilized by the resonance. In addition, N-O bond lengths in complex **1** fall in the ranges found for the corresponding complexes [16,33]. The Pt-S bond length is 2.38012(19) Å and is similar to the values seen in the similar complex {[Pt_2_(SCN)_2_(C_2_H_8_N_2_)_4_](ClO_4_)_4_}_n_ [34]. The S-coordinated thiocyanate group is almost linear, with an N-C-S angle of 177.27(8)°. The lengths of the S-C and C-N in the thiocyanate are 1.6745(7) Å and 1.1613(10) Å, respectively, which suggests the existence of the ligand in the N≡C-S^-^ form. Intramolecular bonding interactions are observed between the oxime ligands where the oxygen in one ligand is bonded to H-C adjacent to the nitrogen in the pyridyl ring of the other ring and vice versa. This contributes to the high coplanarity of the two ligands as the angles N_py_-Pt-N_py_ and N_oxim_-Pt-N_oxime_ are 180.00(3)° and 180.00(2)°, respectively. Moreover, intermolecular bonding interactions are responsible for the crystal packing as the nitrogen of the thiocyanate ligands in one molecule bonds to hydrogen on the pyridyl ring (H_3_) while the oxygen of the oxime establishes another H-bonding with another hydrogen on the pyridyl ring (H_5_) with 3.3847(11) Å and 3.1406(9) Å distances for N—H-C and O—H-C, respectively (Figure 2). Selected interatomic distances and bond angles are shown in Appendix A.

An infinite number of discrete mononuclear molecules extend down the a-axis to form a 1D-chain via H-bonds between the hydrogen atoms of the pyridine ring and the nitrogen atom of the oxime ligand in complex **1**, C—H···N (2.441 Å) (Appendix A). The 1D-chain extended to a 2D-layer by creating strong hydrogen bonds between hydrogen atoms of the pyridine ring of paO ligands in one chain with nitrogen and oxygen atoms of paO ligands in another chain, (2.362–3.068 Å) (Figure 3, Appendix A). Therefore, taking account of all hydrogen bonds in all directions, the supramolecular bonding contacts extend the 2D-layer to a 3D-supramolecular network with fishbone-like structure along the c- axis (Figure 4). Moreover, there are no π… π stacking and metal–metal interactions in complex **1**, as the shortest distance between centroid–centroid in the aromatic ring of paO ligands is 6.485 Å and Pd1–Pd1= 6.486 Å.

On comparing complex **1** with related compounds, Pd(II) complex containing pyridine-2-carbaldehyde-oxime ligand of formula [Pd(C_6_H_5_N_2_O)_2_], **(2)** was recently reported by Moloud Alinaghi et al. [33]. Complex **2** was prepared by reacting pyridine-2-carbaldehyde-oxime ligand with palladium acetate by slow diffusion method to obtain a crystal suitable for an X-ray single crystal. In complex **2**, the coordination number around the palladium atom is four, which indicates that the Pd(II) atom is bonded to two pyridine-2-carbaldehyde-oxime ligands from two imino nitrogens and two pyridine nitrogen atoms of the ligand giving square planar geometry. Our complex **1** was prepared at room temperature by a simple reaction between an aqueous acetonitrile solution of chloroplatinic acid and pyridine-2-carbaldehyde-oxime in the presence of potassium thiocyanate. The Pt(IV) atom in complex **1** is six coordinated by two thiocyanate groups and two pyridine-2-carbaldehyde-oxime ligands giving octahedral geometry.

### 3.2. IR Spectra

The complex **1** and free ligand pyridine-2-carbaldehydeoxime (paOH) were investigated by the FT-IR spectra in the range of 4000–400 cm^−1^ as KBr pellets (Table 1 and Appendix A). The IR spectrum of complex **1** shows one strong band at 2126 cm^−1^ associated with the asymmetric stretch of the thiocyanate group indicating the presence of terminal EO bridging thiocyanate. The IR spectrum of pyridine-2-carbaldehyde-oxime shows a medium band at 3186 cm^−1^ assignable to the stretching vibrations for OH attached to –N=CH group, which disappears in the IR spectrum of complex **1** due to losing the hydrogen of the oxime group. The stretching vibration corresponding to CH bonds (arom. aliph.) appears at 3048, 2925 cm^−1^ in the IR spectrum of **1**. On the other hand, a specific vibration and the ν_CH_ in the IR of free paOH ligand appear at 3186 and 3005 cm^−1^, respectively. The IR spectrum of complex **1** exhibits bands at 1607, 1512, 1429 and 729 cm^−1^ due to ν_C=N,_ ν_C=C,_ δ_CH_ and γ_CH_ of pyridine-2-carbaldehyde-oxime, respectively (Table 1). In comparison to the free paOH ligand, the IR spectrum of complex **1** shifts to higher or lower frequencies, indicating hydrogen bonding formation and supporting paO coordination to the platinum center in complex **1** [35]. The ν_(NO)_ oximate mode is attributed to the middle band at 976 cm^−1^ for free ligand paOH, which increases to 1049 cm^−1^ in complex **1**. This shift to higher frequencies has been discussed, and is consistent with the fact that when oximate N-coordination to platinum atom occurs, there is a higher contribution of N=O to the electronic structure of the oximate group, and thus the (NO) vibration shifts to a higher wavenumber than the free ligand paOH [16,33,35]. The band at 567 cm^−1^ corresponds to υ_(Pt-N)_ in complex **1** and disappears in the IR spectrum of free ligand paOH, confirming the establishment of platinum nitrogen bonds, Table 1**.**

### 3.3. NMR Spectra

The ^1^H NMR spectrum of free pyridine-2-carbaldehyde-oxime ligand (PaOH) shows bands at 7.1–8.5 ppm due to the aromatic protons (H_a_, H_b_, H_c_ and H_d_). These protons are nonequivalent. H_b_ has been split by other hydrogens and appears as a doublet of doublets at 7.16 ppm. H_c_ and H_d_ signals have merged and appear at 7.61–7.85 ppm. H_a_ has resonated as a broad doublet at 8.48 ppm. These protons show up at different frequencies (7.37–9.66 ppm) in the ^1^H NMR spectrum of complex **1** (Appendix A). Furthermore, a singlet can be observed at 7.95 ppm in the free pyridine-2-carbaldehyde-oxime spectrum, which is associated with the imine protons of the ligand that have been shifted to a lower frequency (7.73 ppm) due to the formation of complex 1 (Appendix A). Furthermore, the singlet signal at 11.26 ppm attributed to the proton specific for OH linked to the –N=CH group of the free pyridine-2-carbaldehyde-oxime ligand did not appear in the ^1^H-NMR spectrum of complex **1**, confirming the removal of the hydroxide group in the structure of complex **1**. This assignment was confirmed by comparing with the ^1^H-NMR spectrum of the free pyridine-2-carbaldehyde-oxime which exhibits these peaks at the higher field (ppm). The coordination of pyridine-2- carbaldehyde-oximate ligand to platinum(IV) ion through nitrogen atoms and the creation of hydrogen bonds through the hydroxide group is responsible for the shift in peaks to downfield in the ^1^H-NMR spectrum of complex **1**.

The ^13^C-NMR spectrum of complex **1** exhibits a significant shift of the carbon resonances compared to that of the free pyridine-2-carbaldehyde-oxime ligand (120.0, 126.2, 136.1, 148.95 and 153.6 ppm) because of an electron-density transfer from the ligand to the metal atoms. The ^13^C-NMR spectrum of complex **1** shows six singlet peaks at 119.1, 120.5, 140.5, 140.9, 147.0 and 157.8 ppm due to the six different carbon atoms of pyridine-2-carbaldehyde-oxime ligand. The carbon atom of the thiocyanate group also exhibits a band at 195.9 ppm in the ^13^C-NMR spectrum of complex **1**, as shown in Appendix A. As a result, the NMR spectra of complex **1** are consistent with its structure.

### 3.4. Electronic Absorption and Emission Spectra

The electronic absorption spectrum of pyridine-2-carbaldehyde-oxime (paOH) in DMF shows three bands at 220, 265 and 315 nm (Table 2). The ^1^L_a_←^1^A transition causes the high intensity band at 220 nm, while the ^1^L_b_←^1^A transition causes the broad band at 265 nm [36]. These bands are similar to those found in benzene, and change when substituted [36]. The n-π* transition is responsible for the long wavelength band at 315 nm, which disappears when HCl is added. On the other hand, the spectrum of complex **1** shows absorption bands at 208 and 230 nm corresponding to ^1^L_a_←^1^A and ^1^L_b_←^1^A, respectively. The last band which appears at 278 nm is attributed to π-π* transition (Table 2). The absence of a band due to the n-π* transition, as seen in Appendix A, confirms the coordination of the paO ligand to the platinum center.

Pyridine and its derivatives have been shown to be non-luminescent materials in general [37]. As a result, when excited at 290 nm, the free pyridine-2-carbaldehyde-oxime (paOH) shows no emission bands. On excitation at 290 nm, the fluorescence emission of complex **1** in DMF showed prominent bands at 355 and 415 nm at room temperature (Table 2, Figure 5). The first emission band is caused by the π-π* transition [38], whereas the second band is caused by metal-to-ligand charge transfer (MLCT), as shown in Table 2, Figure 5, [39].

### 3.5. Cytotoxicity of [Pt(SCN)_2_(PaO)_2_] Complex in Human Cancer Cell Lines

Complex **1** was synthesized to investigate its antitumor activities against four human tumor cell lines. In order to calculate the IC_50_ of [Pt(SCN)_2_(PaO)_2_] complex, the percentage of viability of HepG2, HCT_116_, MCF-7, and JK-1 cells after 48 h of treatment with different dosages of complex **1** was calculated (Figure 6). After 48 h incubation of human cell lines treated with the complex of Pt(IV), the IC_50_ values were 19 ± 6, 21 ± 5, 22 ± 6, and 13 ± 3 μM in HepG2, HCT116, MCF-7, and JK cell lines, respectively. The JK treated with complex **1** had the lowest IC_50_ (Table 3). In HepG2, HCT116, MCF-7, and JK cell lines, the IC_50_ of complex **1** ranged from 13–22 μM.

The IC_50_ of cisplatin and oxaliplatin was estimated in non-small cell lung, ovarian, gastric, and colon cancer cell lines (A549, OVCaR-3, SGC-7901, and HCT116). For the aforementioned cells, the IC_50_ cisplatin and oxaliplatin Pt(II) treatment ranges were 5–9 μM and 8–16 μM, respectively. The IC_50_s of cisplatin and oxaliplatin in HCT_116_ were 9 and 16 μM, respectively. In the colon, stomach, ovarian, and lung cancer cell lines, synthesized Pt(IV) complexes were found to be cytotoxic [40]. DNA is blocked by the Pt(II) and Pt(IV) complexes, which hinder DNA replication and transcription, resulting in cell death. Pt complexes may also inhibit cell development by interfering with signaling proteins such as RAS and mitogen-activated protein kinase (MAPK) [40,41].

### 3.6. Morphology of HCT116 and HepG2 Treated with [Pt(SCN)_2_(PaO)_2_] Complex

HCT_116_ and HepG2 cells cultured 2.5 × 10^5^ with the IC_50_ concentration of [Pt(SCN)_2_(PaO)_2_] showed more distorted and unregulated morphology. The number of treated cells was also smaller than the number of control cells, (Figure 7).

### 3.7. Apoptosis and Necrosis of HCT116, HepG2 Treated with [Pt(SCN)_2_(PaO)_2_] Complex

According to our findings, complex **1** at a concentration of 20 μM (which is the dose of the IC50 level, which kills 50% of cells) caused 34% of HCT116 cells to induce apoptosis and 27% of them to induce necrosis. The apoptosis and necrosis folds of HCT116 cells treated with HCT116 cells were 3.2 and 5.5 folds, respectively, in comparison to the untreated cells. In a previous study, cisplatin-treated A549 cells were shown to be in the 15.43% stage of apoptosis. When compared to untreated A549 cells, the apoptotic state of cisplatin-treated A549 cells increased apoptosis by 3.2 when compared to untreated cells [40]. In both treated and untreated HepG2 cells, no evidence of early or late apoptosis was found. In addition, treated HepG2 cells had a 19.7% necrosis rate compared to 14% for untreated cells (Figure 8 and Table 4).

### 3.8. The Cell Cycle Phases of HCT_116_ and HepG2 Treated with [Pt(SCN)_2_(PaO)_2_]

When compared to untreated HCT116 and HepG2 cells, which were arrested in G0/G1 phase (56.4 % and 49.2%), S phase (12.3% and 15.3%), and G2/M (25.8% &28.5 %), HCT116 and HepG2 cells treated with IC_50_ of complex **1** showed a decrease in G0/G1 phase (33% and 44.4%) and an increase in S (21.3% and 29.8%) (Figure 9 and Figure 10 and Table 5). It was mentioned that the cell cycle of A549 treated with cisplatin was arrested to S phase 30.24% (1.5 fold) as compared to untreated cells [40]. When compared to untreated cells, HCT116 and HepG2 cells treated with complex **1** arrested 1.7 and 1.9 folds, respectively, which is identical to the same cell arrested to the S phase in the prior work. This suggests that the replication of DNA treated with complex **1** was stopped in S-phase, which could be due to complex **1** intercalating to DNA [40,42].

### 3.9. Catalytic Activity Study

The catalytic oxidation of benzyl alcohols to aldehydes using complex **1** as a catalyst in aqueous hydrogen peroxide as an oxidant has been studied under solventless conditions, utilizing ultrasonic irradiation. First, to optimize the reaction conditions to get the product selectivity, good yield, and shorter reaction time, several reaction parameters such as solvents, amount of catalyst and reaction temperature were screened using 4-methoxybenzyl alcohol as a model substrate (Figure 2).

In the initial investigation, the oxidation of 4-methoxybenzyl alcohol was carried out with different molar ratios of the oxidant and substrate (Table 6). It is noticeable from the results cited in Table 6 that when we conduct the reaction with the substrate (4-methoxy benzyl alcohol) with an oxidant (H_2_O_2_) ratio [S:O ratio] 1:1.5, in the absence of a catalyst, no product is formed (entry 1). Good results were obtained for a ratio of 1:1.5 with a catalyst of 10 mol% (entry 3). The best outcome was attained when raising the temperature to 60–65 °C (entry 4). Raising the temperature to 80 °C has no effect on the reaction yield (entry 6). The reaction at 5 mol% catalysts with S:O molar ratio of 1:1.5 was not completed until 90 min. (entry 2). Increasing the catalyst mol % to 15 decreases the selectivity in which 4-methoxy benzoic acid was formed with a 4% yield (NMR Yield) (entry 7). Increasing the S:O molar ratio to 1:2 in the presence of 10 mol% catalyst has no effect on reaction yield or selectivity (entry 6). Decreasing the S:O molar ratio to 1:1 under the same reaction conditions decreases the reaction yield and increases reaction time (entry 8). The solvent effect was also tested for the model reaction in water, 1,4-dioxane, and acetonitrile (entries 9–11, respectively), but it was found that the solventless condition was the best for this reaction. Thus, the best reaction conditions that attain the highest selectivity, higher reaction yield, and shorter reaction time are 10 mol% catalyst with S:O molar ratio 1:1.5 at 60–65 °C in solventless conditions under ultrasonic irradiation. Noteworthy, to find the beneficial effect of the ultrasonic irradiation on the above reaction, the above reaction was repeated under the best reaction conditions in the absence of ultrasonic irradiation, with only stirring. It was found that the reaction was completed in 3 h and attained 84% yield. Therefore, ultrasonic waves have an essential role in the above reaction.

The role of ultrasonic irradiation in our catalytic reactions arises from the well-known established mechanism of sonochemistry “cavitations” [43]. Cavitations facilitate hydrogen peroxide to split over the complex **1** catalyst, producing oxidative species (O_2_) [44]. This induces faster oxidation of the targeted alcohol, producing the observed synergistic effect in comparison with silent reaction. It may be noted that sonication using a bath sonicator as in our work is more efficient than other types of sonicators [45].

The promising results of our protocol’s higher conversion and selectivity push us to suggest a tentative mechanism for our reaction. Based on the previous studies [46,47,48] we postulate that the mechanism would involve the reduction of the Pt(IV) complex into Pt(II). First, the Pt complex precatalyst, [Pt^IV^(SCN)_2_(PaO)_2_] (A) reacts with hydrogen peroxide, to form a [Pt^IV^(OSCN)_2_(PaO)_2_] (B) [49]. The active catalyst B reacts with benzyl alcohol derivative to afford a Pt alcoholate (C). Next, the product aldehyde compound and a [Pt^II^(OSCN)_2_(PaOH)_2_] (D) are formed via *β*-hydrogen elimination from species C. The subsequent dehydration of (D) by oxidation with oxygen reproduces species B (Figure 3). The formation of [Pt^II^(OSCN)_2_(PaOH)_2_] (D) was supported by in situ IR monitoring in a separate experiment, isolating D complex via filtration and washing it with water then drying in *vacuo*. The original complex A comprising orange crystals was changed into a brownish solid. The FT-IR of the brownish solid confirms the disappearance of the SCN band due to the formation of OSCN and the appearance of an OH broad band at 3388 cm^−1^ (Figure 11). The scope of our catalytic reaction extended to check other benzyl alcohol derivatives such as 4-bromobenzylalcohol and 4-chlorobenzylalcohol under the above optimized conditions. Only the corresponding aldehydes were obtained in excellent yield, as represented in Figure 12.

## 4. Conclusions

The novel complex **[Pt(SCN)_2_(PaO)_2_] (1)** bearing pyridine-2-carbaldehyde oxime and thiocyanate ligands has been prepared in the present work. Spectral studies and solid-state X-ray crystallography confirmed the coordination mode of the ligands to the platinum and reveal the distorted octahedral geometry around the platinum(IV) center. The tested complex **1** was designed and synthesized to examine its effects on the viability and proliferation of four different cancer lines. Complex **1** was found to be cytotoxic to the colon (HCT_116_), liver (HepG2), breast (MCF-7) and erythroid cells (JK-1). The JK cells experienced the most cytotoxicity. For HCT_116_ cells treated with this Pt(IV) complex, 34 percent and 27.8% of HCT_116_ cells went into apoptosis and necrosis, respectively. Meanwhile, the treated HCT_116_ and HepG2 cells were arrested at S phase to 21.7% and 29.8%, respectively. Finally, this work looked at whether complex **1** may suppress HCT_116_ growth by inducing apoptosis and DNA damage. Thus, complex **1** had promising tumor growth inhibition in the in vitro model. More in vivo experiments are needed for pre-clinical tests and to shed light on the mechanism of drug action. Complex **1** presented a robust catalytic activity. The Pt(IV) complex **1** appeared active under base-free conditions in benzyl alcohol derivatives’ oxidation to the corresponding aldehydes. Furthermore, we found the best selectivity of the catalysts achieved with 10 mol% giving an excellent yield.

## Data Availability

The data presented in this study are available on request from the authors.

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
