# Peer review of "Potential Anticancer Activities and Catalytic Oxidation Efficiency of Platinum(IV) Complex"

_molecules, 2022, doi:10.3390/molecules27144406_

Round 1
Reviewer 1 Report
The manuscript “Potential anticancer activities and catalytic oxidation efficiency of platinum(IV) complex” deals with the synthesis, isolation and characterization of octahedral platinum complex with pyridine-2-carbaldehyde-oxime and thiocyanate ligands. Additionally the behavior of it in the in vitro evaluation of the cytotoxicity against four different cancer cell lines and behavior as catalyst in oxidation of benzyl alcohols to aldehydes were performed.
In general the presented manuscript makes a good impression. However numerous of points to be clarified.
· Line 203. In 1H NMR spectrum of p-anisaldehyde(2a) only 1 of 2 signals of aromatic protons were presented.
· Line 112. The word "about" is a bit confusing when giving the yield.
· It is strange but the NMR spectral data is absent in the experimental part.
· The absence of reaction scheme in the manuscript.
· About synthesis. In the reaction mixture presented: chloroplatinic acid, pyridine-2-carbaldehyde-oxime, potassium thiocyanate, acetonitrile and water. As a result authors isolated complex with deprotonated (!) oxime ligand. I don’t understand in which way oxime proton was removed.
· Unfortunatelly, there are no cif-file present in supporting information
· Ref. [33] International Journal of Hydrogen Energy 2015, 40, (3), 1548-1558 if wrong. You can see DOI 10.1016/j.ijhydene.2014.11.074. There are no information about platinum complex {[Pt2(SCN)2(C2H8N2)4](ClO4)4}n was given there.
· It is very strange that authors not comprised structure of complex 1 with other pyridine-2- carbaldehyde-oximate complexes. For example with article M. Alinaghi et al., J. Molec. Str. 1219 (2020) 128479, https://doi.org/10.1016/j.molstruc.2020.128479
· I think that figures 1a and 3a do not carry any additional information and can be easily removed.
· Also tables 1, 2 and 3 can be removed from manuscript to supporting information.
· Lines 320-322 “The proton of water in solvents is represented as a signal at 3.35 ppm in the 1H-NMR spectrum of complex 1. A prominent singlet band appears in the DMSO protons at 2.5 ppm” must be removed because it’s a common knowledge.
· On fig S1 intensity of peaks is absent.
· Line 328. Error – not “195.9 pm” but “195.9 ppm”.
· In general, the part devoted to NMR is written badly and confusingly. It would be good if the authors compared the spectra of complex 1 and the free ligand. This would give an idea of which signals are being shifted and which signals are characteristic.
· The phrase “structurally characterized spectroscopically” in “Conclutions” is strange.
I believe that the manuscript can be improved and after significant revision can be published in Molecules
Author Response
The manuscript “Potential anticancer activities and catalytic oxidation efficiency of latinum(IV) complex” deals with the synthesis, isolation and characterization of octahedral platinum complex with pyridine-2-carbaldehyde-oxime and thiocyanate ligands. Additionally the behavior of it in the in vitro evaluation of the cytotoxicity against four different cancer cell lines and behavior as catalyst in oxidation of benzyl alcohols to aldehydes were performed.
In general the presented manuscript makes a good impression. However numerous of points to be clarified.
Thank you for your useful comments. All the comments raised were carefully taken into consideration and a list of responses to the comments was also submitted. The manuscript was carefully improved. The corrections are made in red color in the revised manuscript.
- Line 203. In 1H NMR spectrum of p-anisaldehyde(2a) only 1 of 2 signals of aromatic protons were presented.
Response: The second signals of aromatic proton was added, 7.12 (d, J = 8.0 Hz, 2H) .
- Line 112. The word "about" is a bit confusing when giving the yield.
Response: The word about was deleted and the sentence was changed to “The complex 1 was produced with a yield of 85 mg (61%).
- It is strange but the NMR spectral data is absent in the experimental part.
Response: Done. The NMR spectral data were added in the experimental part
- The absence of reaction scheme in the manuscript.
Response: The reaction scheme was added in the manuscript in section 2.3.
- About synthesis. In the reaction mixture presented: chloroplatinic acid, pyridine-2-carbaldehyde-oxime, potassium thiocyanate, acetonitrile and water. As a result authors isolated complex with deprotonated (!) oxime ligand. I don’t understand in which way oxime proton was removed.
Response: the reaction of Pt(IV) with pyridine-2-carbaldehyde-oxime and potassium thiocyanate produces the complex [PtIV(SCN)2(paO)2] with deprotonated oxime ligand due to charge neutrality as Pt (IV) with two anionic thiocyanates and two anionic pyridine-2-carbaldehyde-oxime. This was confirmed by IR, 1HNMR spectra of free ligand and complex 1 which show the absence of stretching band and hydrogen peak of OH group, due to losing the hydrogen of the oxime. Also the X-ray single crystal analysis confirm the deprotonated of hydrogen oxime ligand as the removing of hydrogen decrease the R factor of refinement and give more realty for the structure. This also confirm by literature [16,34].
- Unfortunatelly, there are no cif-file present in supporting information.
Response: The cif file was added in supporting information.
- Ref. [33] International Journal of Hydrogen Energy 2015, 40, (3), 1548-1558 if wrong. You can see DOI 10.1016/j.ijhydene.2014.11.074. There are no information about platinum complex {[Pt2(SCN)2(C2H8N2)4](ClO4)4}n was given there.
Response: sorry for this mistake, the reference was wrong. The corrected reference was added giving information with complex {[Pt2(SCN)2(C2H8N2)4](ClO4)4}n. the correct reference is: Ozawa, Y.; Kima, M.; Toriumi, K.; A one-dimensional platinum mixedvalence complex with bridging thiocyanate S atoms: [[PtII(en)2](m-SCN)-[PtIV(en)2](m-SCN)](ClO4)4 (en is ethane-1,2-diamine. Acta Crystallographica 2013, C69, (2), 146–149.
- It is very strange that authors not comprised structure of complex 1 with other pyridine-2- carbaldehyde-oximate complexes. For example with article M. Alinaghi et al., J. Molec. Str. 1219 (2020) 128479, https://doi.org/10.1016/j.molstruc.2020.128479.
Response: Done, the structure of complex 1 was compared with other pyridine-2- carbaldehyde-oximate complexes and the reference was cited [34], see the last section 3.1.
I think that figures 1a and 3a do not carry any additional information and can be easily removed.
Response; Done, the figures were removed.
- Also tables 1, 2 and 3 can be removed from manuscript to supporting information.
Response; Done, the Tables were moved to supporting information
- Lines 320-322 “The proton of water in solvents is represented as a signal at 3.35 ppm in the 1H-NMR spectrum of complex 1. A prominent singlet band appears in the DMSO protons at 2.5 ppm” must be removed because it’s a common knowledge.
Response; Done, the sentence was removed.
- On fig S1 intensity of peaks is absent.
Response Done, the intensity of peaks was added on Fig.S1.
- Line 328. Error – not “195.9 pm” but “195.9 ppm”.
Response; Done, the sword was corrected.
- In general, the part devoted to NMR is written badly and confusingly. It would be good if the authors compared the spectra of complex 1 and the free ligand. This would give an idea of which signals are being shifted and which signals are characteristic.
Response: Done, the NMR spectra of the complex 1 was compared with the free ligand of pyridine-2-carbaldehyde-oxime. The section 3.3 of NMR spectra was rewritten.
- The phrase “structurally characterized spectroscopically” in “Conclutions” is strange.
Response; The phrase was removed and changed to “The novel complex [Pt(SCN)2(PaO)2] (1) bearing pyridine-2-carbaldehyde oxime and thiocyanate ligands has been prepared in the present work”.
I believe that the manuscript can be improved and after significant revision can be published in Molecules
-Thank you again for your comments.
Reviewer 2 Report
The manuscript submitted by M. M. El-bendary and coauthors is devoted to the synthesis and investigation of the biological and catalytic activities of the new Pt(IV) complex featuring pyridine-2-carbaldehyde-oxime and thiocyanate ligands. This is an interesting report with valuable findings both from the viewpoint of medicinal chemistry and catalysis. It can be published after minor revision according to the following comments.
1. The characteristics of the benzaldehydes resulting from the catalytic oxidation of different substituted benzyl alcohols (section 2.8) should be completely removed from the manuscript or transferred to the Supporting Information (all of them are the well-known compounds, therefore, there is no need to characterize them).
2. The values of IC50 for the complex under consideration must be given with the standard deviation.
3. The percentages of apoptotic cells must be presented with the same accuracy (compare, for example, 34%, 27.8%, and 4.57%). This also refers to the necrosis rate. Furthermore, the total percentage of apoptotic cells in the control experiment with HCT116 cell culture seems to be too high (over 10%). What is the reason?
4. The revised manuscript should be thoroughly spell checked and grammar checked before submission. Consider, for example, the following cases:
pentagonal ring --> five-membered ring;
bridging hiocyanate --> bridging thiocyanate;
was confirmed by consulting the 1H-NMR spectrum of paO --> was confirmed by comparting with the 1H-NMR spectrum of paO;
Complex 1 was synthesized to investigate their antitumor activities --> Complex 1 was synthesized to investigate its antitumor activities.
Author Response
The manuscript submitted by M. M. El-bendary and coauthors is devoted to the synthesis and investigation of the biological and catalytic activities of the new Pt(IV) complex featuring pyridine-2-carbaldehyde-oxime and thiocyanate ligands. This is an interesting report with valuable findings both from the viewpoint of medicinal chemistry and catalysis. It can be published after minor revision according to the following comments.
Thank you for your useful comments. All the comments raised were carefully taken into consideration and a list of responses to the comments was also submitted. The manuscript was carefully improved. The corrections are made in red color in the revised manuscript
- The characteristics of the benzaldehydes resulting from the catalytic oxidation of different substituted benzyl alcohols (section 2.8) should be completely removed from the manuscript or transferred to the Supporting Information (all of them are the well-known compounds, therefore, there is no need to characterize them).
Response: Done, the section 2.8 was completely moved to Supporting Information
- The values of IC50 for the complex under consideration must be given with the standard deviation.
Response: Done, the values of IC50 for the complex 1 were given with the standard deviation and added in Table 3 and text.
- The percentages of apoptotic cells must be presented with the same accuracy (compare, for example, 34%, 27.8%, and 4.57%). This also refers to the necrosis rate. Furthermore, the total percentage of apoptotic cells in the control experiment with HCT116 cell culture seems to be too high (over 10%). What is the reason?
Response: Due to the generation of HCT 116 cells (passage of cells), the apoptotic state was 10.5% in the control experiment (passage of cells). The number of passage of this cells used was 12. In our work, complex 1-treated HCT 116 cells were contrasted with control cells. The section 3.7 was rewritten and improved.
- The revised manuscript should be thoroughly spell checked and grammar checked before submission. Consider, for example, the following cases:
pentagonal ring --> five-membered ring;
bridging hiocyanate --> bridging thiocyanate;
was confirmed by consulting the 1H-NMR spectrum of paO --> was confirmed by comparting with the 1H-NMR spectrum of paO;
Complex 1 was synthesized to investigate their antitumor activities --> Complex 1 was synthesized to investigate its antitumor activities.
Response: Done, the words were corrected. The manuscript was checked carefully for spell and grammar
Reviewer 3 Report
Huge number of Pt(IV) complexes have been already synthesized in the literature to investigate as potential anticancer agents. Here an additional one was synthesized in a simple reaction and characterized. The biological results do not show extremely high cytotoxic effects on the cell-lines tested.
Also the catalytic effect of the complex was investigated in the oxidation reaction of benzyl alcohols in the presence of an oxidant. Based on the results a tentative scheme of the mechanism for the reaction is presented in the manuscript. This scheme, however, is not exactly supported.
All together, I found that the manuscript presents results collected mainly in some kind of routine-like experiments. As many experiments as could be done, the authors did those, but explanation of a real research goal, compact concept would be beneficial.
Author Response
Thank you for your useful comments. All the comments raised were carefully taken into consideration and a list of responses to the comments was also submitted. The manuscript was carefully improved. The corrections are made in red color in the revised manuscript
Huge number of Pt(IV) complexes have been already synthesized in the literature to investigate as potential anticancer agents. Here an additional one was synthesized in a simple reaction and characterized. The biological results do not show extremely high cytotoxic effects on the cell-lines tested.
Response: The IC50 for complex 1 in 4 different cell lines (HepG2, HCT116, MCF-7, and JK cell lines) ranged from 13-22.3 mM. (Table 3). In the prior study, the IC50 for complex 1 was comparable or almost high (not excessively high). The IC50 ranges for the treatments with cisplatin and oxaliplatin on Pt(II) were found to be 5.45-9.05 mM and 8.29-16.09 mM, respectively. It will be advised to conduct additional research in the future on the toxicity of complex 1 and compare it to cisplatin in rats with cancer that has been induced.
Also the catalytic effect of the complex was investigated in the oxidation reaction of benzyl alcohols in the presence of an oxidant. Based on the results a tentative scheme of the mechanism for the reaction is presented in the manuscript. This scheme, however, is not exactly supported.
Response: Thank you for reviewer comment, we already supposed the mechanism based on the IR resulted for the used catalyst as represented in Fig. 11, in which we found disappearance of thiocyanate band, which suggest that reaction between thiocyanate and hydrogen peroxide to form OSCN, also, appearance of band due to OH support the catalyst intermediate (D) consequently support the reduction of PtIV into PtII..
All together, I found that the manuscript presents results collected mainly in some kind of routine-like experiments. As many experiments as could be done, the authors did those, but explanation of a real research goal, compact concept would be beneficial.
Response: Thank you for reviewer comment, we rewrite the introduction part to declare the main target of our manuscript in which, most of benzyl alcohol derivatives acquire an economic value when oxidized to aldehydes, either because aldehydes are known intermediates to high-value components, or because they have a very high market value themselves, being widely used in perfume, cosmetics and food industry [X]. The effective catalytic conversion of benzyl alcohol derivatives requires the study of the relation between the substrate features and the catalytic behavior at molecular level. The correlation between the catalyst structural chemistry and the characteristics of the reactant in a specific environment allows to establish rules for a successful catalysts design. Indeed, among the factors that strongly influenced the catalytic activity, the substrate effect can play a crucial role. Pt complexes are an environmental-friendly alternative for the oxidation of alcohols in base-medium, although the main product is often the corresponding monoacid and not the desired aldehyde.
[X] Xu, C.; Arancon, R.A.D.; Labidi, J.; Luque, R.., Lignin depolymerisation strategies: towards valuable chemicals and fuels. Chemical Society Review 2014, 43, (22) 7485–7500.
Round 2
Reviewer 1 Report
The authors have significantly revised and improved the manuscript. I think now it can be published in Molecules without further revision.
Author Response
Comments and Suggestions for Authors: The authors have significantly revised and improved the manuscript. I think now it can be published in Molecules without further revision.-Response: Thank you again for your comments and for accepting our manuscript.
Reviewer 3 Report
Accepting the revised version for publication is suggested.
Author Response
Comments and Suggestions for Authors: Accepting the revised version for publication is suggested.
Response: Thank you for your useful comments. The English language and style spell check were carefully improved by native English speaker. The manuscript was carefully improved. The corrections are made in red color in the revised manuscript